# Migration confers winter survival benefits in a partially migratory songbird

**Daniel Zúñiga[1,2], Yann Gager[1,2], Hanna Kokko[3], Adam Michael Fudickar[1,2,4], Andreas Schmidt[1], Beat Naef-Daenzer[5], Martin Wikelski[1,2], Jesko Partecke[1,2]\***

[1]Max Planck Institute for Ornithology, Radolfzell, Germany; [2]Department of Biology, University of Konstanz, Konstanz, Germany; [3]Department of Evolutionary Biology and Environmental Studies, University of Zurich, Zurich, Switzerland; [4]Environmental Resilience Institute, Indiana University, Bloomington, United States; [5]Swiss Ornithological Institute, Sempach, Switzerland

**Abstract** To evolve and to be maintained, seasonal migration, despite its risks, has to yield fitness benefits compared with year-round residency. Empirical data supporting this prediction have remained elusive in the bird literature. To test fitness related benefits of migration, we studied a partial migratory population of European blackbirds (*Turdus merula*) over 7 years. Using a combination of capture-mark-recapture and radio telemetry, we compared survival probabilities between migrants and residents estimated by multi-event survival models, showing that migrant blackbirds had 16% higher probability to survive the winter compared to residents. A subsequent modelling exercise revealed that residents should have 61.25% higher breeding success than migrants, to outweigh the survival costs of residency. Our results support theoretical models that migration should confer survival benefits to evolve, and thus provide empirical evidence to understand the evolution and maintenance of migration.
DOI: https://doi.org/10.7554/eLife.28123.001

## Introduction

The adaptive function of migration has often been hypothesized to be a selective advantage to escape adverse situations caused by seasonal fluctuations of food resources or environmental conditions. This seasonality may impose considerable constraints to life, particularly during the winter season. Seasonal migration allows animals to cope with temporal environmental fluctuations by moving between geographically distant habitats (*Fryxell and Sinclair, 1988*). Given that much of our planet offers seasonally varying resources, it is not surprising that migration has evolved repeatedly in many taxa (*Chapman et al., 2011*). Theoretical research on the evolution of migration (*Lundberg, 1987*; *1988*; *Taylor and Norris, 2007*; *Griswold et al., 2010*; *Kokko, 2011*; *Shaw and Levin, 2011*; *Shaw and Couzin, 2013*) has yielded a key prediction: migration should offer either survival or breeding benefits compared to residency. In anadromous fish, for example, individuals migrate between freshwater and ocean habitats. Recent comparisons of migrant and resident steelheads (*Oncorhynchus mykiss*) found that female migrants have higher fecundity than females that remain in fresh water streams (*Satterthwaite et al., 2009*; *Hodge et al., 2014*, *2016*). Similarly, the noctuid moth (*Autographa gamma*) performs a multi-generational migration which confer substantial reproductive benefits by allowing a lineage to spread to multiple sites (*Chapman et al., 2012*). Regarding survival benefits, individuals of a fresh water fish (*Rutilus rutilus)*, increase their survival during the winter by migrating from lakes to streams to avoid predation risks (*Skov et al., 2013*).

In birds, seasonal migration has often been argued to bring about survival benefits, as it allows individuals to avoid inhospitable conditions during the non-breeding season, while the same region can offer abundant resources during the breeding season (*Lack, 1954*). Species exhibiting

**\*For correspondence:**
partecke@orn.mpg.de

**Competing interests:** The authors declare that no competing interests exist.

**eLife digest** Winter is one of the most challenging seasons for many animals. Cold temperatures, bad weather, short days, long nights and a shortage of food can impose a deadly threat. To avoid these inhospitable conditions, some animals migrate to warmer climes during the winter. These animals include many songbirds, which return to the same habitat in the following spring because it offers abundant resources that are thought to help them to breed more successfully. Yet, migration itself can be risky, and there is little empirical data on the survival benefits of migration in songbirds.

Zúñiga et al. tested whether songbirds that migrate are actually more likely to survive the winter than those that do not migrate. The study focused on a population of European blackbirds over a period of seven years. Some of these birds migrated from the breeding grounds in Germany to their wintering sites in southern Europe, whereas others remained all year at the breeding grounds.

Zúñiga et al. found that migrant blackbirds were 16% more likely to survive the winter than the residents. Yet during the summer, there was no difference in survival between the two groups. This raised the question, if migration confers survival benefits, why do some birds do not migrate at all?

Theory predicts that those birds that do not migrate should have some reproductive benefit instead. This makes sense given that birds which remain at the breeding grounds would have access to prime breeding sites which are limited. Using mathematical modelling, Zúñiga et al. estimated how much of reproductive benefits the residents would need to outweigh their greater risk of not surviving the winter. The model predicted that residents should have at least 61.25% higher breeding success than migrants.

The results provide empirical evidence to help scientist understand how migration evolves and becomes maintained in animal population. Future studies are now needed to confirm the estimated breeding success of both groups. Also, because many songbirds are threatened by human activity during migration and at their overwintering sites, future studies to understand how, where and why migratory songbirds die will be important to direct the conservation efforts to protect migratory species.

DOI: https://doi.org/10.7554/eLife.28123.002

polymorphisms in migratory behavior provide an excellent opportunity to test predictions of fitness components. In partially migratory species, some individuals migrate while others remain as year-round residents, thereby allowing for between-group comparisons within the same population. Theory predicts that if residency enhances breeding success in territorial birds, then there should be a corresponding benefit to migrants; higher survival over non-breeding season is a clear, but empirically understudied, possibility (*Lundberg, 1987*; *Kokko, 2011*). Despite the extensive research done on bird migration, there is limited empirical evidence regarding its fitness benefits, as data on fitness-related variation in migratory strategies are logistically difficult to collect in the field. Despite logistical challenges, studies on European robins (*Erithacus rubecula)* and American dippers (*Cinclus mexicanus)* report that migrants have lower survival and reproductive success than residents (*Adriaensen and Dhondt, 1990*; *Gillis et al., 2008*; *Green et al., 2015*). Further, a recent study comparing fitness measures of resident and migrant cormorants (*Phalacrocorax aristotelis*) reported higher reproductive success in residents compared to migrants (*Grist et al., 2017*).

We studied a partially migratory population of European blackbirds (*Turdus merula*) (*Figure 1*) to test whether migration confers survival benefits during the winter. The migrants of our population overwinter, on average, 800 km west-southwest from the breeding grounds (*Fudickar and Partecke, 2012*) (*Figure 2a and b*) and the majority of migrants are females (*Fudickar et al., 2013*). We fitted multi-event survival models using presence-absence data obtained by capture-mark-recapture and radio-telemetry of 192 resident and 70 migrant free-living blackbirds over the course of seven years. These models account for variation in re-encounter probabilities in relation to space, time and behaviour of the birds.

We compared the survival probabilities between residents and migrants during two different seasons: summer (mean start date: March 2 ± 14.5 days - mean end date: November 2 ± 7.4 days) and winter (mean start date: November 3 ± 7.4 days - mean end date: March 1 ± 14.5 days). Based on

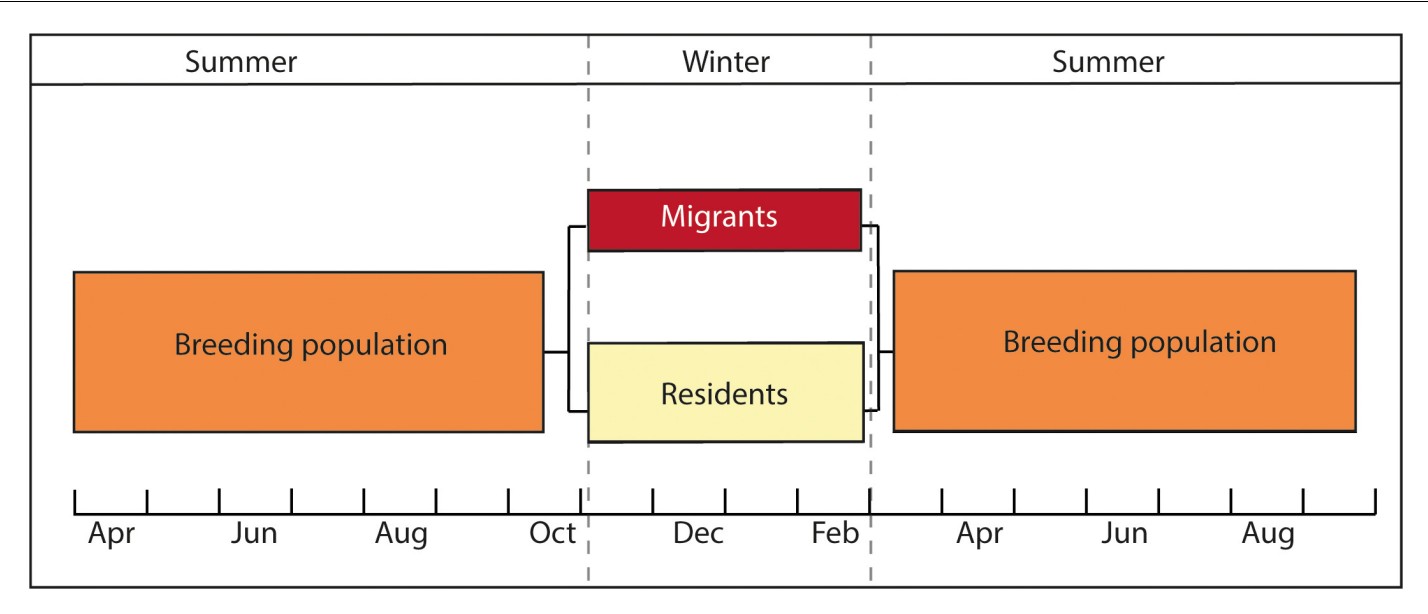

**Figure 1.** Diagram representing the partial migratory system of the population under study. Migrants and residents of a breeding population of European blackbirds are in sympatry during the summer months (March - October). During the wintering months (November – March) migrants and residents overwinter in different habitats.

DOI: https://doi.org/10.7554/eLife.28123.003

theoretical models of partial migration in birds (*Kokko, 2011*), which assume that residency offers reproductive benefits (access to better breeding territories) and that migration should confer survival benefits for at least some individuals if the winter conditions at the breeding ground are harsh, we predicted that migrants should have higher survival probabilities during the winter period, whereas summer survival might not differ between migrants and residents.

## Results and discussion

We found that winter mortality is an important determinant of lifespan, as blackbirds had lower probability to survive the winter ($\Phi = 0.60$; 95% CI = 0.55–0.66) than the summer season ($\Phi = 0.89$; 95% CI = 0.82–0.94) (*Table 1*, model 3) despite the shorter duration of the former season. This result strongly supports the hypothesis that migration confers survival benefits compared with residency as an alternative strategy.

There was no difference between juveniles and adults in survival probability within a season. Juveniles ($\Phi = 0.89$; 95% CI = 0.80–0.94; model 4 *Table 1*) have similar probability to survive the summer as adults ($\Phi = 0.90$; 95% CI = 0.83–0.94; model 4 *Table 1*). During winter, juveniles also have a comparable probability ($\Phi = 0.59$; 95% CI = 0.49–0.68; model 4 *Table 1*) to survive as adults ($\Phi = 0.61$; 95% CI = 0.55–0.67; model 4 *Table 1*).

In line with our predictions, migratory European blackbirds had higher winter survival rates than resident blackbirds. The best model (model 1, *Table 1*) estimated markedly higher winter survival for migrants ($\Phi = 0.73$; 95% confidence intervals (CI) = 0.62–0.81, *Figure 3*) than for residents ($\Phi = 0.57$; 95% CI = 0.50–0.63, *Figure 3*), taking into account the lower detection probability for migrants (P=0.19; 95% CI = 0.13–0.26, *Figure 3*) compared to residents (P=0.74; 95% CI = 0.69–0.78). Our second model, which included sex and had modest support (model 2, delta AICc = 0.95, *Table 1*), predicted that migrants have higher winter survival probability than residents, which was also predicted by model 1. Sex differences were not substantial (during summer: male residents $\Phi = 0.90$; 95% CI = 0.83–0.95; female residents $\Phi = 0.89$; 95% CI = 0.89–0.94; male migrants $\Phi = 0.95$; 95% CI = 0.89–0.98, female migrants $\Phi = 0.94$; 95% CI = 0.87–0.97; during winter: male residents $\Phi = 0.58$; 95% CI = 0.51–0.65; female residents $\Phi = 0.53$; 95% CI = 0.44–0.62; male migrants $\Phi = 0.75$; 95% CI = 0.63–0.84, and female migrants $\Phi = 0.71$; 95% CI = 0.60–0.81;

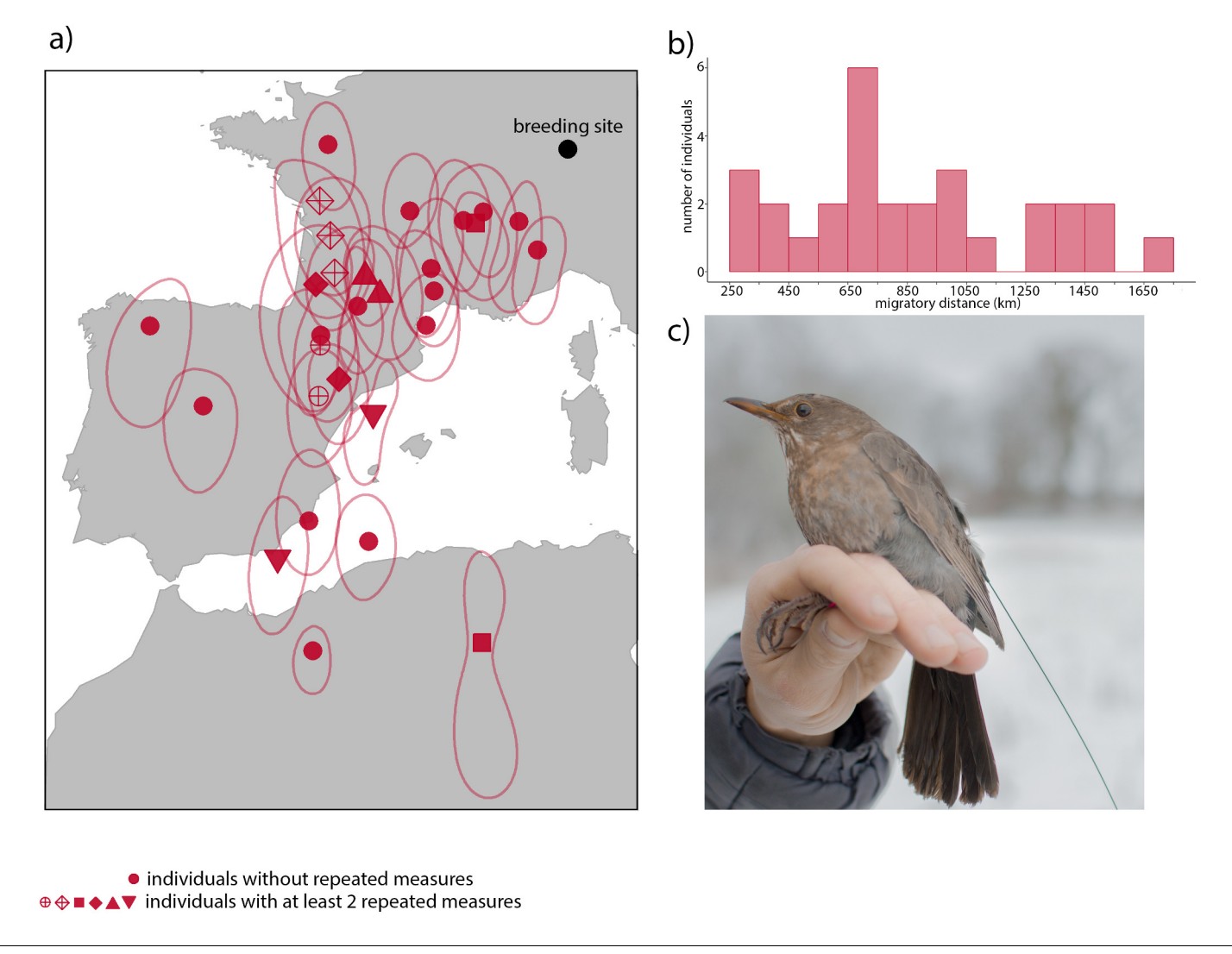

**Figure 2.** Overwintering locations and migratory distance of migrant European blackbirds (*Turdus merula)* between 2009 and 2014. (**A**) Mean overwintering locations (red symbols) and 25% kernel utilization distribution (red lines) of 22 blackbirds were calculated using the light-level data acquired by geolocators during the wintering months (November – February). Raw light level data were processed using the R package 'GeoLight'(**Lisovski and Hahn, 2012**) and Kernel utilization distributions were calculated to estimate the error of each location. Filled red circles represent 16 individuals with one single measurement. The other symbols represent six individuals with at least two repeated measurements in different years. (**B**) Histogram of the migratory distance of migrants. (**C**) Female radio-tagged blackbird.

DOI: https://doi.org/10.7554/eLife.28123.004

The following source data is available for figure 2:

**Source data 1.** Zip file contains five files: 'locations_data.csv'; 'kud2009.

DOI: https://doi.org/10.7554/eLife.28123.005

**Source data 2.** Lat and long and distance to the breeding grounds (km) to the 29 overwintering locations of European blackbirds (2009–2014) used to generate histogram of *Figure 2* panel b.

DOI: https://doi.org/10.7554/eLife.28123.006

detection probability was lower for migrants (P=0.19; 95% CI = 0.13–0.26) than for residents (P=0.74; 95% CI = 0.69–0.78)). It is reassuring that both models 1 and 2 agree on the importance of residency vs. migration in winter, while we refrain from making strong statements regarding the effect of sex, given that *Burnham and Anderson, 2002* advise against considering inferior models competitive in cases like our model 2 (delta AIC within about 0–2 units of the best model, the

**Table 1.** Models examining effects of various covariates (Season, migratory strategy, sex, age) on survival (Φ) and detection probabilities (P) of a partially migratory population of European blackbirds between 2009 and 2016. All models were compared to the base model using Akaike's Information Criterion (AICc), Delta AICc, and changes in model deviance (Dev).

| Model | Number of parameters | QAICc | Delta AICc (Δi) | Weights (ϖi) | Deviance |
|---|---|---|---|---|---|
| (1) Φ [season + migr.].P[migr] | 5 | 1408.3 | 0.00 | 0.59 | 1398.2 |
| (2) Φ [season + migr + sex.].P[migr] | 6 | 1409.2 | 0.95 | 0.36 | 1397.1 |
| (3) Φ [season].P[migr] | 4 | 1414.3 | 6.04 | 0.02 | 1406.2 |
| (4) Φ [season + juv + ad.].P[migr] | 5 | 1416.1 | 7.79 | 0.02 | 1406.0 |
| (5) Φ [migr].P[migr] | 4 | 1447.1 | 38.82 | 0.01 | 1439.0 |
| (6) Φ [.].P[migr] | 3 | 1448.2 | 39.98 | 0.00 | 1442.2 |
| (7) Φ [sex].P[migr] | 4 | 1449.9 | 41.6 | 0.00 | 1441.9 |
| (8) Φ [season + migr].P[season] | 5 | 1504.0 | 95.75 | 0.00 | 1493.9 |
| (9) Φ [season].P[season] | 5 | 1504.2 | 95.89 | 0.00 | 1494.1 |
| (10) Φ [season + sex + migr].P[season] | 6 | 1504.9 | 96.60 | 0.00 | 1492.7 |
| (11) Φ [season + sex].P[.] | 3 | 1530.3 | 122.02 | 0.00 | 1524.3 |
| (12) Φ [season + sex]. P[.] | 4 | 1530.6 | 122.4322 | 0.00 | 1522.6 |
| (13) Φ [season + migr].P[.] | 4 | 1531.0 | 122.7 | 0.00 | 1522.9 |
| (14) Φ [season + juv + ad.]. P[.] | 4 | 1523.2 | 123.0 | 0.00 | 1523.2 |
| (15) Φ [season + sex + migr].P[.] | 5 | 1532.0 | 123.7 | 0.00 | 1521.9 |
| (16) Φ [.].P[Season] | 3 | 1528.9 | 126.7 | 0.00 | 1528.9 |

DOI: https://doi.org/10.7554/eLife.28123.007

difference being caused by one parameter added to the best model and the log-likelihood essentially unchanged).

Our findings support the theoretical predictions that migration yields survival benefits during the winter. In addition, our results provide an explanation for the maintenance of the migrant phenotype in the partially migratory population of European blackbirds that we studied. Residency is predicted to provide reproductive benefits given that year-round occupancy provides, for example, advantages in establishing breeding territories (*Kokko, 2011*). The two phenotypes can persist as evolutionary stable strategies (ESS) due to frequency dependent selection if the overall fitness of migrants and residents is equal (*Lundberg, 1987*). Given the lack of data on the reproductive performance of migrants and residents in our present study, we estimated how much the reproductive performance of residents should be to compensate the survival benefits of migration. If we assume migrant and resident winter survival to be 0.73 and 0.57 respectively, and summer survival of 0.89 for both strategies, then we can estimate the expected number of reproductive attempts for a migrant as $0.73 + \frac{0.73 \times 0.89}{1 - 0.73 \times 0.89} = 2.58$, and $0.57 + \frac{0.57 \times 0.89}{1 - 0.57 \times 0.89} = 1.60$ for residents. Therefore, the expected lifetime number of reproductive seasons is 61.25% higher for migrants than for residents due to higher survival of the former. This calculation assumes that the first breeding season requires one overwintering to be completed successfully, while all other events require an additional surviving sequence of summer followed by winter before a new breeding event can happen. The format for this assumption is $s_1 s_2 / (1 - s_1 s_2)$ which is the solution for the series $s_1 s_2 + (s_1 s_2)^2 + (s_1 s_2)^3 + \ldots$, ($s_1$ corresponds to winter survival probability and $s_2$ to summer survival probability), each subsequent term requiring one sequential survival event through one summer and one winter season.

We conclude from this calculation that the reproductive performance of residents would have to be 61.25% higher than in migrants to achieve equal fitness of the alternative strategies. Such benefits could come about from prior residency effects (either occupying a better territory or avoiding floater status), combined with a longer time spent in the breeding habitat which can make multiple nesting or re-nesting (in case of failure) more likely. Considering that blackbirds are a multi-brood species (2–3 broods a year), it could be possible that residents gain a 61.25% higher breeding

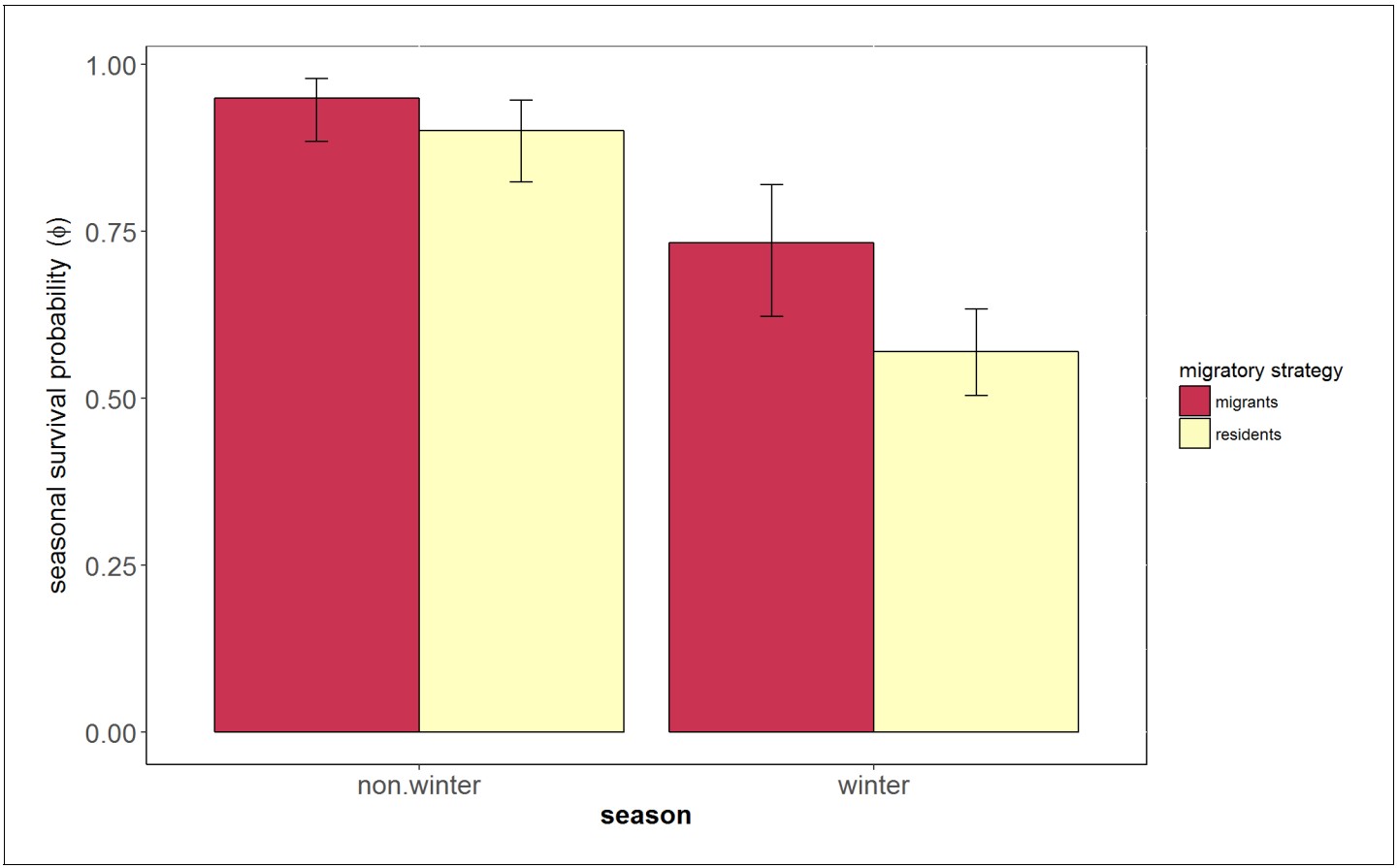

**Figure 3.** Seasonal survival probability of migrants and residents European blackbirds. Survival probability (Φ) and 95% confidence intervals of migrants (red) and residents (yellow) birds estimated using the best ranked multievent capture - mark recapture model (Φ [season +migr].P[migr]). Detection probability (P) was estimated as 0.74 for residents and 0.19 for migrants. 262 birds were included in this analysis (192 were classified as residents and 70 were classified as migrants).

DOI: https://doi.org/10.7554/eLife.28123.008

The following source data is available for figure 3:

**Source data 1.** Results of the best ranked model (Model 1).

DOI: https://doi.org/10.7554/eLife.28123.009

success compared to migrants. Future studies need to confirm these calculations. If resident breeding success is higher than 61.25%, then the fitness of migrants will be lower than the fitness of residents and migration would be a conditional strategy operating under frequency-dependent selection. For conditional migration strategies, some intrinsic phenotypic characteristics (sex, age, dominance) result in a need to adopt a strategy that might yield overall lower fitness than what residents on average achieve, but it is the better choice to optimize individual fitness (*Lundberg, 1987*). To distinguish between these two alternatives, data of reproductive success for this species are needed (note that comparisons within existing studies, such as *Grist et al., 2017* on cormorants, do not incorporate all the processes we have envisaged above).

It is also plausible that year-to-year variation of winter environmental conditions at the breeding grounds play a role shaping the incidence of migration versus residency over time. For instance, during a harsh and long winter, the survival of residents might be lower compared to a mild and short winter. If fewer residents survive an unusually harsh winter and establish breeding territories during the subsequent breeding season, many high-quality territories would remain vacant for migrants to take advantage of after arrival in the spring. Furthermore, if residents that do survive harsh winters begin the breeding season in poor condition, then physically dominant migrants could successfully take-over prime territories from residents. Under this scenario, the prior residency effect would not

be acting at full strength (*Drent et al., 2003*; *Jahn et al., 2010*; *Kokko, 2011*) and migrants would gain breeding benefits.

We found no evidence for sex differences in survival (though some ambiguity remains, as a moderately supported model two includes sex as an explanatory variable — note that the best model does not). This raises the question: why are females more likely to migrate than males in the study population (*Fudickar et al., 2013*)? We can think of two potential reasons for this observation: either there is differential survival, or differential breeding success for each sex. Regarding survival, one line of thinking is to argue that residency is more dangerous for females than for males, because overwintering blackbirds form foraging flocks and an individual's access to food is related to its position within the flock hierarchy (*Lundberg and Schwabl, 1983*). Within these flocks, females are subordinate to males (*Lundberg and Schwabl, 1983*; *Lundberg, 1985*). Therefore, females would be predicted to suffer higher mortality if they remain as residents during winter, when food is limited, than if they migrate. However, our data do not align perfectly with this interpretation: if overwinter survival during residency was a strong factor driving sex differences in migratory strategy, we ought to have seen lower winter survival in resident females than in males, but this was not the case. The other possible explanation relies on differential breeding success between sexes. It is conceivable that resident males enjoy priority access to prime territories as soon as the breeding season starts. However, it should always be remembered that females, too, may benefit from better territories, thus an early presence may be beneficial for them as well (*Creighton, 2001*; *Kokko et al., 2006*; *Kokko, 2011*; *Snow, 1956*). It would be important to understand exactly how territory acquisition differs between males and females, especially because earlier data from the same geographical area have shown that reproductive success of migrant and resident blackbirds is sex-dependent (*Schwabl, 1983*) such that male residents have higher reproductive success than male migrants, while female residents and migrants have similar reproductive success. Understanding the mechanisms of territory acquisition could help explain why fewer males migrate: if frequency-dependency penalizes late-arriving males whereas late breeding females are not severely penalized, then the same magnitude of survival differences will favor a larger migratory population within females than within males.

In our study, we excluded 11 birds that migrated during the winter and 11 that switched strategies between years, as we considered these sample sizes to be too small for detailed inferences. Departures during the winter usually occurred during periods of cold temperatures and snow accumulation (*Fudickar et al., 2013*). Extreme weather conditions and low food availability might trigger these movements during winter. Future, more extensive studies could conceivably determine lifetime fitness of these strategies. By examining the fitness benefits conferred by migration, our study is able to provide strong support for the hypothesis that migration confers winter survival benefits.

Our methodology, which allows comparisons between classes that differ greatly in detectability, can hopefully also shed light on systems where benefits and risks of migration are shared by all individuals of a population, many of which are threatened by risks along their migratory flyways (*Wilcove and Wikelski, 2008*). Further understanding of how, where and why migratory animals die will illuminate the path to direct conservation efforts to protect migratory species.

## Materials and methods

### Capturing and tagging

A total of 469 blackbirds were captured and tagged in a mixed deciduous/coniferous forest in southern Germany (N 47° 47′, E 9° 2′) during spring and summer over seven consecutive years (2009–2016). Sex and age were determined using plumage differences (*Svensson, 1992*). Juvenile birds were sexed using DNA-based sex identification (*Griffiths et al., 1998*). To this end, a blood sample (50 µl) was collected and stored in Queen's Lysis buffer (*Seutin et al., 1991*).

Each bird was equipped with a radio transmitter in combination with (i) a light-level geolocator (Mk 10S, and Mk 12S ≤ 1.2 g; British Antarctic Survey, Cambridge, UK) during 2009–2011, or (ii) light-level geolocator (Intigeo-P65 ≤1.2 g Migrate Technology, Cambridge, UK) during 2012–2013 or (iii) a Pinpoint GPS logger (≤2 g; Biotrack Ltd, Dorset, UK) during 2014. Birds tagged during 2015, however, were equipped just with a radio transmitter (mean weight ±SD: 1.94 g ± 0.12). Radio transmitters were provided in 2009–2012 and 2014–2015 by Sparrow Systems, Fisher, IL, USA and in

2014 by The Swiss Ornithological Institute, Sempach, Switzerland. The total weight of the devices carried by the birds was (mean ±SD) 3.9 g ± 0.19 in 2009–2011; 3.3 g ± 0.20 in 2012; 4.15 g ± 0.11 in 2013; 4.13 g ± 0.11 in 2014. The total weight of the tracking devices was less than 5% of the body weight of the birds in each year of the study. The life span of the battery was at least 12 months. The tags were attached by means of leg-loop harnesses.

## Data collection

We collected presence–absence data at regular intervals through a manual and/or an automated radio telemetry system. Manual radio tracking was carried out twice per week using a handheld three element Yagi antenna (AF Antronics, Inc., Urbana, IL, USA) and AR 8200 MKIII handheld receiver (AOR U.S.A., Inc., Torrance, CA, USA) or a handheld H antenna (Andreas Wagener Telemetry Systems, Köln, Germany) and a Yaesu VR 500 handheld receiver (Vertex Standard USA, Cypress, CA, USA).

The automated radio telemetry system consisted of 4 to 6 stationary automated receivers (ARU) (Sparrow Systems, Fisher, IL, USA) deployed at the study site. Each receiver was connected to an H antenna (ATS, Isanti, MN, USA) and was able to search for up to 16 different radio frequencies every 60 s.

## Migratory strategy determination

The migratory strategy of each bird was assigned based on the presence–absence data. Birds were classified as migrants if they departed at night (determined by ARUs) from the study site during the autumn (September-November). All migrants departed between 19 September and 12 November (mean departure date: 16 October). Migrants arrived during spring between 17 February and 25 March (mean arrival date: 14 March). Birds were classified as residents if they remained present and alive at the study site at least until 31 November of each year. Individuals that had left the study area were searched using a Cessna airplane fitted with two H antennas and two Biotrack receivers (Lotek Wireless Inc., Newmarket, ON, Canada) and a vehicle carrying a telescopic mast (6 m height) and three-element Yagi antenna (Vargarda Radio, Vargarda, Sweden). Out of 469 birds, 158 were excluded because their migratory strategy could not be determined before 31 November due to various reasons (technical failure of the tracking devices, dispersal from the core study area or mortality). In 9 out of the 158 excluded birds, we found a radio tag with a broken harness and in 16 cases we found the tag but no signs of predation nor malfunction were evident. We concluded that 27 birds were predated (predation signs e.g. the carcass and/or feathers were found near the radio transmitter). In 106 cases, we do not know the fate of the birds. Forty-nine of the 106 birds with unknown fates were juveniles. In blackbirds, as in many other altricial bird species, post-fledgling mortality is high (*Naef-Daenzer and Grüebler, 2016*) and fledglings can disperse several kilometres (*Paradis et al., 1998*). Eleven birds that departed from the breeding grounds during the winter and 11 birds that switched strategies across years were excluded from the analysis. Finally, we excluded 27 juveniles from the analysis because we could not determine the sex due to poor quality of the blood sample. Conversely 262 birds were classified during the autumn and were included in the survival analysis. Out of 262, 192 were classified as residents (69 females: 52 adults and 17 juveniles; 123 males: 96 adults and 27 juveniles) and 70 birds were classified as migrants (45 females: 28 adults and 17 juveniles; 25 males: 17 adults and 8 juveniles).

## Data preparation

To estimate seasonal survival probabilities, each calendar year was divided into two operationally defined 'seasons': summer and winter. Summer was defined as the period of time between the date of the first spring arrival of a migrant bird and the date of the last departure in the fall (mean start date: March 2 ± 14.5 days, mean end date November 2 ± 7.4 days). To define the start of the first summer season in 2009, the date of the very first capture (April 23rd) was used. Winter, in turn, corresponded to the period of time between the date of the last departure in the fall and the date of first arrival the subsequent spring (mean start date: November 3 ± 7.4 days, mean end date: March 1 ± 14.5 days). Based on the presence–absence data, we generated a matrix of 15 columns, each corresponding to one respective season (summer 2009, followed by winter 2009–2010, followed by summer 2010, etc.) and 262 rows (one for each individual). Additional columns containing the

covariates sex (males and females), age at capture (juveniles and adults) and migratory status (migrants and residents) were added to the matrix.

## Statistical analysis

We implemented multi-event models using the software E-SURGE 1.9.0 (*Choquet et al., 2009*). These models belong to the family of hidden Markov models. They assume that the individuals in a population independently transition between a finite set of $N$ states (e.g. presence, absence) through a finite number of sampling occasions. Since the capture regime is imperfect, there is uncertainty in presence or absence of each individual. Multi-event models account for this uncertainty (*Pradel, 2005*). They allow a simultaneous estimation of the probability of survival ($\Phi$) of a group of individuals and its detection probability (P). Detection probability (P) is a decisive parameter because it directly influences the survival estimates and in natural populations often is less than 1. Failing to account for this parameter can lead to incorrect conclusions in capture mark-recapture analyses (*Gimenez et al., 2008*).

We used a model selection procedure to evaluate the performance of 16 candidate models that included the effects of sex, age at capture, migratory strategy and season (*Table 1*). Model performance was evaluated using the Akaike Information Criterion corrected for small samples (AICc). Delta AICc ($\Delta$ AICc) was calculated and the models ranked based on this value.

## Geolocator data processing

After recapture, Mk 10S and Mk 12S geolocators were processed in the following way: Raw data were corrected for clock drift using Bastrak (British Antarctic Survey). After visually inspecting light values of Mk 10S geolocators, a light level threshold of 16 was identified. In 2010, we reduced the stalk length of Mk 12 s geolocators, resulting in interference from feathers with light censors. Due to interference at sunrise and sunset, we found that a light threshold of 2 was most reliable for Mk 12 s geolocators. Individual sun elevation angles were calculated using all dates that an individual was known to be on the breeding grounds. Transitions were calculated using TransEdit2 (British Antarctic Survey) and anomalous transitions were rejected. Latitude and longitude were calculated using Locator (British Antarctic Survey) following *Tøttrup et al. (2012)*. Intigeo-P65 geolocators were processed in the following way: Transitions were calculated using IntiProc v.1.01 (Migrate Technology Ltd). A threshold of 16 was used and anomalous transitions were discarded. Transition data were imported and analyzed in R using GeoLight Package (*Lisovski and Hahn, 2012*). The 'in-habitat calibration' was used to calculate individual sun elevation angles. Locations (latitude and longitude) were estimated using the function 'Coord' of the GeoLight Package.

# Acknowledgements

We want to thank Olivier Gimenez, Guillaume Souchay and Jean-Dominique Lebreton for their advice provided in the statistical analysis and guidance in the use of E-SURGE. Brian Cusack gave comments to an early version of the manuscript. DZ, YG and AMF were supported by the International Max Planck Research for Organismal Biology.

# Additional information

### Funding

| Funder | Grant reference number | Author |
|---|---|---|
| Max-Planck-Gesellschaft | Open-access funding | Daniel Zúñiga<br>Yann Gager<br>Adam Michael Fudickar<br>Andreas Schmidt<br>Martin Wikelski<br>Jesko Partecke |

The funders had no role in study design, data collection and interpretation, or the decision to submit the work for publication.

## Author contributions

Daniel Zúñiga, Conceptualization, Data curation, Formal analysis, Methodology, Writing—original draft, Writing—review and editing; Yann Gager, Formal analysis, Writing—review and editing; Hanna Kokko, Formal analysis, Writing—review and editing, Conceptualization of the model to estimate breeding success; Adam Michael Fudickar, Conceptualization, Data curation, Methodology, Writing—review and editing; Andreas Schmidt, Methodology, Capture and tagging of birds. Collection of a substantial amount of the presence absence radio telemetry data. Management of the automatic radio telemetry system; Beat Naef-Daenzer, Methodology, Writing—review and editing, development of radio transmitters used in 2013; Martin Wikelski, Supervision, Funding acquisition, Methodology, Writing—review and editing; Jesko Partecke, Conceptualization, Supervision, Methodology, Project administration, Writing—review and editing

## Author ORCIDs

Daniel Zúñiga (iD) http://orcid.org/0000-0001-7198-7242
Jesko Partecke (iD) http://orcid.org/0000-0002-9526-8514

## Ethics

Animal experimentation: All the experimental procedures were performed in accordance with the German regulation on animal experimentation. The experimental protocol was approved by the Ethical Committee of Baden-Wüttemberg under permit 35-9185.81/G-09/08 and 35-9185.81/G-13/29.

## Decision letter and Author response

Decision letter https://doi.org/10.7554/eLife.28123.014
Author response https://doi.org/10.7554/eLife.28123.015

# Additional files

## Supplementary files

• Source data 1. Raw presence data derived from capture-recapture and radio telemetry data of 262 European blackbirds used for the survival analysis.
DOI: https://doi.org/10.7554/eLife.28123.010

• Source data 2. Presence-absence matrix during winter and summer of 262 European blackbirds between 2009–2016. This matrix was built with *Source data 1* .
DOI: https://doi.org/10.7554/eLife.28123.011

• Source data 3. Zip file containing the necessary files to run the a session of the E-surge software (estimation of multi-events survival models). More information on E-surge see *Gimenez et al., 2014*. Download for free here: https://www.cefe.cnrs.fr/fr/ressources/films/34-french/recherche/bc/bbp/264-logiciels File called "seasonalfms.mod", contains the actual session to be opened in E-surge. This session contains all the models fitted (*Table 1*). File called "GEPAT-blackbirds.pat", contains the GEPAT file. Input file for E-surge necessary to estimate the transition matrix between presence absences events. File called "matrix.fm-s.males.females.non.winter.winter2.txt" contains the raw presence absence data in E-surge format used to fit the multi-events survival models.
DOI: https://doi.org/10.7554/eLife.28123.012

• Transparent reporting form
DOI: https://doi.org/10.7554/eLife.28123.013

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
