## [Decision Letter]

Thank you for submitting your article "Migration confers winter survival benefits in a partially migratory songbird" for consideration by *eLife*. Your article has been reviewed by four peer reviewers, the Reviewing editor and an external advisor, and the evaluation has been overseen by Barbara Helm as the Reviewing Editor and Ian Baldwin as the Senior Editor. The following individuals involved in review of your submission have agreed to reveal their identity: Per Lundberg (Reviewer #1); Jason Chapman (Reviewer #3).

The reviewers have discussed the reviews with one another. All reviewers agree that this is an important paper in migration ecology. The data set, collected over 7 years, will make an important contribution to this field in its struggles to provide data on the costs and benefits of animal movements.

The Reviewing Editor has drafted this decision to help you prepare a revised submission.

Summary:

This manuscript provides exciting new data on a much debated topic in migration biology: why do some animals move while others remain residential? Researchers working with different systems have tried to answer this question in different ways, and theoretical models have been constructed to identify the factors that could maintain co-existence of migrants and residents within populations. In the field of bird migration, researchers found it difficult to quantify costs and benefits encountered by migrant and resident fractions during different parts of the annual cycle. The current manuscript reports data from a long-term (7-year) study of partially migratory European Blackbirds that were tagged with radio-transmitters and automatically recorded. The researchers quantified actual migration distances, estimated from geolocators. They then demonstrated increased winter survival probability for migrants compared to residents. Finally, using modelling, they estimated the increase in productivity required of migrants if the system was to remain stable. Unfortunately, empirical measures of productivity were not available for comparison.

Overall, this manuscript is well written, interesting, timely and will help resolve the debate in the field. We have the following suggestions to improve the manuscript.

Essential revisions:

1) Partial migration: as a reviewer put it, the subject of the paper is a bird, and as written the paper is rather bird-centric. Hence, to be relevant to migration research more broadly, the study should also consider other animal taxa for which relevant research has been carried out (as well as be inclusive of earlier avian studies, for example on American Dippers and White Storks). Reviewers give constructive ideas for greater coverage, in particular for fish, for which full parameterized life history information is available, and for insects, where migration can comprise several generations. A broader perspective will also lead to a more circumspect conceptual framework, in particular where the manuscript currently is narrowly prescriptive (e.g. regarding fitness: "superior winter survival is necessary to compensate.…"). When developing a more robust framework, please also include precise predictions for the sexes and age groups which are analytical categories in your model.

2) Materials and methods section: Reviewers and an external advisor have expressed some concern about the high equipment failure or mortality suffered in the study. They cite the authors as having to exclude 288 of 572 subjects because they "died before their migratory strategy could be determined or tracking failed for technical causes". Reviewers worry about this very substantial proportion of all deployments, asking for more information about these cases and possible effects on results. In particular, an external advisor wondered how the 4 different tracking technologies (tag mass from 1.2–2.6g) may have differentially impacted individuals. On a related issue, please report in greater detail how geolocator data were analysed and consider performing model averaging, rather than selecting best models.

3) Apparent errors: Please check very carefully the inconsistencies reported by the reviewers, for example regarding numbers of data points versus numbers of birds, and make sure to provide all results, including those on effects of age and all cited Tables.

It would be appropriate to acknowledge the very first paper on the theory of the topic:

Lundberg, 1987. Partial bird migration and evolutionarily stable strategies.

lease be meticulous with statements about fitness trade-offs. One of the recurring statements in the paper is that seasonal migration will occur only if it confers a survival-related benefit during winter. Migration can of course also evolve/persist in species because it confers a breeding-related benefit. Anadromy (migration to/from the ocean) in fishes is a good example (Gross, 1987; Hendry et al., 2004; Hodge et al., 2016). The Introduction could do more to discuss the importance of breeding-related trade-offs. Also, migration can, in theory, confer a survival-related benefit during any time of year. Keep in mind that a life history can persist in a population so long as it occasionally confers a fitness advantage. The relative costs and benefits of migration/residency probably vary by year and with external conditions (e.g. environment).

Another suggestion is to either tone back comments about the paucity of parameterized fitness models, or qualify the information gap. Here are a just a few quick examples of parameterized life history models involving partially migratory fishes: Satterthwaite et al., 2009; Satterthwaite et al., 2010; Hodge et al., 2014; Morita et al., 2014.

Although it's probably too late, or outside the scope of the study, to quantify the breeding-related costs of migration (breeding-related benefits of residency), the topic warrants more consideration. Perhaps the authors could further elaborate on this point and discuss how/why incidence of migration/residency might fluctuate in time.

Abstract: The point should arguably be less about human "understanding" and more about persistence of a trait in birds.

Abstract: Be careful with use of "this". It's not really clear what "this core idea" is-that data have remained elusive? That seasonal migration is difficult to understand?

Abstract: This statement should be qualified.

Introduction section: The word "strategy" has specific implications in the context of life history. See Gross, (1987).

Introduction section: To the contrary, a migration evolved (and persists) in Pacific salmonids whereby fish migrate to sea despite a net survival cost.

Introduction section: See general comments.

Introduction section: Can probably delete the "a" in front of "polymorphism". Good point about testing predictions.

Results and Discussion section: See general comments about interpreting AIC rankings. Where is Table 1?

Results and Discussion section: This prediction should be mentioned in the Introduction.

Results and Discussion section: Suggest offering more explanation in the Methods and fewer equations in the Results and Discussion.

Results and Discussion section: Exactly. This is a more appropriate way to frame the statement.

Results and Discussion section: Rather, did not detect survival differences between sexes.

Results and Discussion section: "Punishes" doesn't seem appropriate. "Penalizes" would be better. Better still is to take the time of describing a fitness cost.

Results and Discussion section: The fact alone is not compelling evidence for maintenance of two phenotypes. The survival advantage favors only one phenotype.

Subsection “Migratory strategy determination”: Did all migrants spend approximately the same time away from the breeding grounds?

Subsection “Migratory strategy determination”: Although it was probably easier to exclude the mixed-strategy birds, they might present an interesting test case. Is it possible that the "switchers" benefit from higher survival than residents and greater breeding success than migrants?

I'm not a statistician, but as I understand it, inclusion of variables in best-supported models is not in itself evidence that the variables are significant. Also, the world is shifting from a "best model" approach to a model-averaged approached (see Arnold, 2010; Doherty et al., 2012). It would not take much to calculate model-averaged parameter estimates and the results would be more robust for it. Finally, the authors could report cumulative AIC weights and/or new β values for each variable. Β values with CIs would of course inform readers about the statistical significance of season and migratory form as predictors.

The subject of the paper is a bird, and as written the paper is rather bird-centric (or at least vertebrate oriented) in its approach, with the reference list dominated by vertebrate studies. To be relevant to the entire field of animal migration research, the authors could make some efforts to widen the scope. For example, –in the Introduction section, the authors state that migration can only evolve if migrants have a net survival benefit compared to residents, but it seems to me that this is a vertebrate phenomenon. What about multi-generational migrants (such as insects) where multiple migrations and breeding bouts occur throughout the year (including the "winter"). In a partially migrant insect system for example, migration could evolve if migrant lineages produce more progeny over the course of several generations of winter-breeding than resident lineages which survive winter by diapausing (and therefore not breeding). I would suggest broadening the introduction so that it covers all taxa will increase the impact of this paper.

Results and Discussion section: What is known about the reproductive performance of residents, is there any evidence that it is around 60% higher than migrants as predicted? Unless it is significantly higher, then migration should be favoured and expected to outcompete residency, but the number of migrants and residents in your study indicates that residency outnumbers migration by 2:1.

References: The formatting is all over the place here.

Figure 2: The legend indicates 22 birds were tracked, but plot A appears to have 28 points, while plot B appears to have data from 26 birds.

Figure 3: The bars are yellow not orange

Results and Discussion section: I'm not sure how these confidence intervals relate to the probability values, as the detection probability for migrants is given as 0.19 but has confidence intervals of 0.69 – 0.78.

Materials and methods section: Repeated text here.

Introduction: "is necessary" I find this an over-generalisation of points that might apply to the system the authors have on mind. Many alternatives are possible, for example that later-returning migrants arrive in better state, with high reward, as proposed for example by Drent et al., 2003).

---

## [Author Response]

Summary:This manuscript provides exciting new data on a much debated topic in migration biology: why do some animals move while others remain residential? Researchers working with different systems have tried to answer this question in different ways, and theoretical models have been constructed to identify the factors that could maintain co-existence of migrants and residents within populations. In the field of bird migration, researchers found it difficult to quantify costs and benefits encountered by migrant and resident fractions during different parts of the annual cycle. The current manuscript reports data from a long-term (7-year) study of partially migratory European Blackbirds that were tagged with radio-transmitters and automatically recorded. The researchers quantified actual migration distances, estimated from geolocators. They then demonstrated increased winter survival probability for migrants compared to residents. Finally, using modelling, they estimated the increase in productivity required of migrants if the system was to remain stable. Unfortunately, empirical measures of productivity were not available for comparison.Overall, this manuscript is well written, interesting, timely and will help resolve the debate in the field. We have the following suggestions to improve the manuscript.

Thank you for the kind words. We have implemented improvements as detailed below.

Essential revisions:1) Partial migration: as a reviewer put it, the subject of the paper is a bird, and as written the paper is rather bird-centric. Hence, to be relevant to migration research more broadly, the study should also consider other animal taxa for which relevant research has been carried out (as well as be inclusive of earlier avian studies, for example on American Dippers and White Storks). Reviewers give constructive ideas for greater coverage, in particular for fish, for which full parameterized life history information is available, and for insects, where migration can comprise several generations. A broader perspective will also lead to a more circumspect conceptual framework, in particular where the manuscript currently is narrowly prescriptive (e.g. regarding fitness: "superior winter survival is necessary to compensate.…"). When developing a more robust framework, please also include precise predictions for the sexes and age groups which are analytical categories in your model.

We broadened the Introduction section in the revised version of the manuscrip; we now cite more extensively, taking examples from fish and insect literature in addition to birds (for which we also cite a new paper, published in 2017, in cormorants). We now steer clear of narrowly prescriptive a priori statements, see the beginning of the Introduction section. Regarding sexes and age groups: making precise a priori predictions would, unfortunately, make us fall into the same trap that we are trying to avoid now – i.e. being too narrowly system-specific, as it is not quite clear which sex or age group in our case should be a priori more prone to migrate. However, we have analysed these effects for completeness sake and present the results accordingly (Results and Discussion section); but these being rather system-specific effects, they are not the main focus of the manuscript and the results indicate we should remain open-minded about the effects.

2) Materials and methods section: Reviewers and an external advisor have expressed some concern about the high equipment failure or mortality suffered in the study. They cite the authors as having to exclude 288 of 572 subjects because they "died before their migratory strategy could be determined or tracking failed for technical causes". Reviewers worry about this very substantial proportion of all deployments, asking for more information about these cases and possible effects on results. In particular, an external advisor wondered how the 4 different tracking technologies (tag mass from 1.2–2.6g) may have differentially impacted individuals. On a related issue, please report in greater detail how geolocator data were analysed and consider performing model averaging, rather than selecting best models.

These are very important comments. Upon rereading our first submitted version of the manuscript, we realised that what we had written about the total number of birds used in this study was perhaps misleading. In the previous version we included in the total N, birds that were captured but not radio-tagged (103 birds), giving a misleadingly high impression of failure.

We now write more clearly about the numbers reported in the previous version of the MS and added a more detailed information about the other remaining numbers of birds which we had to exclude from the analyses. The total amount of tagged birds was 469. From this total we had to exclude 158 birds because their migratory strategy could not be determined (technical failure, mortality), 22 birds which switch strategies and 27 juveniles which their sex could not be determined. Out of the 158 birds that were excluded, 40.5% (64 birds) were juveniles. In blackbirds, as in many other altricial bird species, post-fledgling mortality is high (Naef-Daenzer, 2016) and fledglings can disperse several kilometres (Paradis, 1998). As a consequence, we lost signals of some birds because they gradually moved out of our study site. In the current version we report this more clearly, detailing the number of cases where we found a radio transmitter and a carcass of a bird, the number of cases we found a transmitter with a broken harness, and the number of cases where we found the transmitter but could not determine the apparent cause of death or transmitter malfunction:

“Out of 469 birds, 158 were excluded because their migratory strategy could not be determined before 31 November due to various reasons (technical failure of the tracking devices, dispersal from the core study area or mortality). In 9 out of the 158 excluded birds, we found a radio tag with a broken harness and in 16 cases we found the tag but no signs of predation nor malfunction were evident. We concluded that 27 birds were predated (predation signs e.g. the carcass and or feathers were found near the radio transmitter). In 106 cases, we do not know the fate of the birds. Forty-nine of the 106 birds with unknown fates were juveniles. In blackbirds, as in many other altricial bird species, post-fledgling mortality is high (Naef-Daenzer and Grüebler, 2016) and fledglings can disperse several kilometres (Paradis et al., 1998). Eleven birds that departed from the breeding grounds during the winter and 11 birds that switched strategies across years were excluded from the analysis. Finally, we excluded 27 juveniles from the analysis because we could not determine the sex due to poor quality of the blood sample. Conversely 262 birds were classified during the autumn and were included in the survival analysis. Out of 262, 192 were classified as residents (69 females: 52 adults and 17 juveniles; 123 males: 96 adults and 27 juveniles) and 70 birds were classified as migrants (45 females: 28 adults and 17 juveniles; 25 males: 17 adults and 8 juveniles).” Subsection “Migratory strategy determination”.

Regarding the weight of the tracking devices, we now include more precise information about the total weight of the different tracking technologies:

“Birds tagged during 2015, however, were equipped just with a radio transmitter (mean weight ± SD: 1.94 g ± 0.12). Radio transmitters were provided in 2009-2012 and 2014-2015 by Sparrow Systems, Fisher, IL, USA and in 2014 by The Swiss Ornithological Institute, Sempach, Switzerland. The total weight of the devices carried by the birds was (mean ± SD) 3.9 g ± 0.19 in 2009 – 2011; 3.3 g ± 0.20 in 2012; 4.15 g ± 0.11 in 2013; 4.13 g ± 0.11 in 2014. The total weight of the tracking devices was less than 5% of the body weight of the birds in each year of the study. The life span of the battery was at least 12 months. The tags were attached by means of leg-loop harnesses.” Subsection “Capturing and tagging.

Regarding the geolocator data processing we added a more detailed explanation of how the data were processed:

“Geolocator data processing:After recapture, Mk 10S and Mk 12S geolocators were processed in the following way: Raw data were corrected for clock drift using Bastrak (British Antarctic Survey). After visually inspecting light values of Mk 10S geolocators, a light level threshold of 16 was identified. In 2010 we reduced the stalk length of Mk 12 s geolocators, resulting in interference from feathers with light censors. Due to interference at sunrise and sunset, we found that a light threshold of 2 was most reliable for Mk 12s geolocators. Individual sun elevation angles were calculated using all dates that an individual was known to be on the breeding grounds. Transitions were calculated using TransEdit2 (British Antarctic Survey) and anomalous transitions were rejected. Latitude and longitude were calculated using Locator (British Antarctic Survey) following Tottrup et al., 2011. Intigeo-P65 geolocators were processed in the following way: Transitions were calculated using IntiProc v.1.01 (Migrate Technology Ltd). A threshold of 16 was used and anomalous transitions were discarded. Transition data were imported and analyzed in R using GeoLight Package (Lisovski and Hahn, 2012). The “in-habitat Calibration” was used to calculate individual sun elevation angles. Locations (Latitude and Longitude) were estimated using the function “Coord” of the GeoLight Package.” Subsection “Geolocator data processing”.

3) Apparent errors: Please check very carefully the inconsistencies reported by the reviewers, for example regarding numbers of data points versus numbers of birds, and make sure to provide all results, including those on effects of age and all cited Tables.

We carefully checked inconsistencies in Figure 2. The fact that we reported 22 birds but there are 29 points in Figure 2 results from 6 birds offering repeated measurements in different years. In the current version of the MS we represent these cases with different symbols in the figure. Regarding the inconsistency in Figure 2 in the previous version: there are fewer observations because the x axis was truncated due to a mistake in the R code. In the current version we solved this issue. In the previous version the table with the different models was provided in the supplementary material. In the current version the table is provided in the manuscript.

It would be appropriate to acknowledge the very first paper on the theory of the topic: Lundberg, 1987. Partial bird migration and evolutionarily stable strategies.

Thank you for the comment we added the Reference in the Introduction section.

Please be meticulous with statements about fitness trade-offs. One of the recurring statements in the paper is that seasonal migration will occur only if it confers a survival-related benefit during winter. Migration can of course also evolve/persist in species because it confers a breeding-related benefit. Anadromy (migration to/from the ocean) in fishes is a good example (Gross, 1987; Hendry et al., 2004; Hodge et al., 2016). The Introduction could do more to discuss the importance of breeding-related trade-offs. Also, migration can, in theory, confer a survival-related benefit during any time of year. Keep in mind that a life history can persist in a population so long as it occasionally confers a fitness advantage. The relative costs and benefits of migration/residency probably vary by year and with external conditions (e.g. environment).

The reviewer is absolutely right pointing out the fitness trade-offs in relation to migration. We have now changed the text accordingly, i.e. by replacing the sentence:

“Theoretical research on the evolution of migration (3–8) has yielded a key prediction: migration cannot evolve unless migrants have a net survival benefit compared to year-round residents.”

with “Theoretical research on the evolution of migration (Lundberg, 1987, 1988; Taylor and Norris, 2007; Griswold et al., 2010; Kokko, 2011; Shaw and Levin, 2011; Shaw and Couzin, 2013) has yielded a key prediction: migration should offer either survival or breeding benefits compared to residency.” Introduction section.

Furthermore, we now discuss empirical evidence of fitness benefits described in the fish and insect literature:

“Recent comparisons of migrant and resident steelheads (Oncorhynchus mykiss) found that female migrants have higher fecundity than females that remain in fresh water streams (Satterthwaite et al., 2009; Hodge et al., 2014, 2016). Similarly, the noctuid moth (Autographa γ) performs a multi-generational migration which confer substantial reproductive benefits by allowing a lineage to spread to multiple sites (Chapman et al., 2012). Regarding survival benefits, individuals of a fresh water fish (Rutilus rutilus), increase their survival during the winter by migrating from lakes to streams to avoid predation risks (Skov et al., 2013).” Introduction section.

Another suggestion is to either tone back comments about the paucity of parameterized fitness models, or qualify the information gap. Here are a just a few quick examples of parameterized life history models involving partially migratory fishes: Satterthwaite et al., 2009; Satterthwaite et al., 2010; Hodge et al., 2014; Morita et al., 2014.

We thank the reviewer for this suggestion. In this version we toned back the comments about the lack of parameterized fitness models in relation to migration and, as mentioned previously, we included, in the revised version, citations from empirical and theoretical studies in the literature on fishes and insects.

Although it's probably too late, or outside the scope of the study, to quantify the breeding-related costs of migration (breeding-related benefits of residency), the topic warrants more consideration. Perhaps the authors could further elaborate on this point and discuss how/why incidence of migration/residency might fluctuate in time.

The reviewer is right that it is outside the scope of the study to quantify breeding-related costs and benefits of migration vs residency. In the current revised manuscript, we discuss the fact that we do not have this type of data in more detail and that, indeed, this was one of the reasons why we performed the model exercise to see what benefits would be necessary to outweigh the costs (Results and Discussion section). Furthermore, we very much appreciate the reviewer´s comment regarding the fluctuation of migration vs residency over time. We added a paragraph in the Results and Discussion section about this topic:

“It is also plausible that year-to-year variation of winter environmental conditions at the breeding grounds play a role shaping the incidence of migration versus residency over time. For instance, during a harsh and long winter, the survival of residents might be lower compared to a mild and short winter. If fewer residents survive an unusually harsh winter and establish breeding territories during the subsequent breeding season, many high-quality territories would remain vacant for migrants to take advantage of after arrival in the spring. Furthermore, if residents that do survive harsh winters begin the breeding season in poor condition, then physically dominant migrants could successfully take-over prime territories from residents. Under this scenario, the prior residency effect would not be acting at full strength (Drent et al., 2003; Jahn et al., 2010; Kokko, 2011) and migrants would gain breeding benefits.”

Abstract: The point should arguably be less about human "understanding" and more about persistence of a trait in birds.

We agree, and we changed the sentence “seasonal migration is difficult to understand unless migration, despite its risks, yields survival benefits compared with year-round residency.”

to read “To evolve and to be maintained, seasonal migration, despite its risks, has to yield fitness benefits compared with year-round residency.”. Abstract.

Abstract: Be careful with use of "this". It's not really clear what "this core idea" is-that data have remained elusive? That seasonal migration is difficult to understand?

We changed the sentence “To test this core idea behind evolutionary theories of migration we studied a partial migratory population of European blackbirds (Turdus merula) over 7 years”.

for “To test fitness related benefits of migration, we studied a partial migratory population of European blackbirds (Turdus merula) over 7 years”. Abstract.

Abstract: This statement should be qualified.

We added the following sentence “showing that migrant blackbirds had 16% higher probability to survive the winter compared to residents.” Abstract.

Introduction section: The word "strategy" has specific implications in the context of life history. See Gross, (1987).

We appreciate the reviewer´s comment. We deleted the word “strategy” in the Introduction section.

Introduction section: To the contrary, a migration evolved (and persists) in Pacific salmonids whereby fish migrate to sea despite a net survival cost.

We changed the sentence: “Theoretical research on the evolution of migration (3–8) has yielded a key prediction: migration cannot evolve unless migrants have a net survival benefit compared to year-round residents”

to “Theoretical research on the evolution of migration (Lundberg, 1987, 1988; Taylor and Norris, 2007; Griswold et al., 2010; Kokko, 2011; Shaw and Levin, 2011; Shaw and Couzin, 2013) has yielded a key prediction: migration should offer either survival or breeding benefits compared to residency.” Introduction section.

Introduction section: Can probably delete the "a" in front of "polymorphism". Good point about testing predictions.

We deleted preposition “a” in the Introduction section.

Results and Discussion section: See general comments about interpreting AIC rankings. Where is Table 1?

We deleted the sentence “The robustness of these findings is indicated by the fact that both two best-supported models in an AIC approach used season and migratory strategy as explanatory variables (Supplementary file 1, model 1 & 2)”. We are sorry for the confusion about the position of Table 1. In the previous version Table 1 was in the supplementary material. In current revised manuscript we decided to include Table 1 in the main manuscript.

Results and Discussion section: This prediction should be mentioned in the Introduction.

We added the following paragraph in the Introduction section:

“Based on theoretical models of partial migration in birds (Kokko, 2011), which assume that residency offers reproductive benefits (access to better breeding territories) and that migration should confer survival benefits for at least some individuals if the winter conditions at the breeding ground are harsh, we predicted that migrants should have higher survival probabilities during the winter period, whereas summer survival might not differ between migrants and residents.”

Results and Discussion section: Suggest offering more explanation in the Methods and fewer equations in the Results and Discussion.

Thanks for the reviewer comment, however, we believe that explaining the equation in parallel to our calculations helps the reader to follow easier the exercise and line of thought.

Results and Discussion section: Exactly. This is a more appropriate way to frame the statement.

Thank you.

Results and Discussion section: Rather, did not detect survival differences between sexes.

We changed the sentence “We did not detect significant sex differences in survival.” to “We found no evidence for sex differences in survival (though some ambiguity remains, as a moderately supported model 2 includes sex as an explanatory variable — note that the best model does not).” Results and Discussion section.

Results and Discussion section: "Punishes" doesn't seem appropriate. "Penalizes" would be better. Better still is to take the time of describing a fitness cost.

We changed the word “punishes” for “penalizes” in the Results and Discussion section.

Results and Discussion section: The fact alone is not compelling evidence for maintenance of two phenotypes. The survival advantage favors only one phenotype.

We appreciate the reviewer´s comment. The original sentence “The fact that migration offers survival benefits for European blackbirds during winter provides a compelling explanation for the maintenance of the two phenotypes (migrants and residents) in this population.”

now reads “Our findings support the theoretical predictions that migration yields survival benefits during winter. In addition, our results provide an explanation for the maintenance of the migrant phenotype in the partially migratory population of European blackbirds that we studied.” Results and Discussion section.

Subsection “Migratory strategy determination”: Did all migrants spend approximately the same time away from the breeding grounds?

There was variation in the departure and arrival dates. We included a couple of sentences describing this fact. “All migrants departed between 19 September and 12 November (mean departure date: 16 October). Migrants arrived during spring between 17 February and 25 March (mean arrival date: 14 March.)”subsection “Migratory strategy determination”.

Subsection “Migratory strategy determination”: Although it was probably easier to exclude the mixed-strategy birds, they might present an interesting test case. Is it possible that the "switchers" benefit from higher survival than residents and greater breeding success than migrants?

This is an interesting point and it might be the case that “switchers” have a higher fitness than the other two strategies. However, our sample sizes do not permit testing this hypothesis in a meaningful manner. Our data also show that there is quite some within-individual consistency in the strategies, thus switchers are rare. One can speculate, e.g., if switching offered the best of both worlds in a flexible manner, one would expect this to be more commonly observed (however we cannot rule out that a low incidence of switching is due to a low year-to-year variation in environmental conditions during winter throughout our study) — but given sample size limitations, we would like to avoid making strong statements. In the revised version we write:

“In our study, we excluded 11 birds that migrated during the winter and 11 that switched strategies between years, as we considered these sample sizes to be too small for detailed inferences. Departures during the winter usually occurred during periods of cold temperatures and snow accumulation (Fudickar et al., 2013). Extreme weather conditions and low food availability might trigger these facultative migratory movements. Future, more extensive studies could conceivably determine lifetime fitness of these strategies. By examining the fitness benefits conferred by migration, our study is able to provide strong support for the hypothesis that migration confers winter survival benefits.” Results and Discussion section.

I'm not a statistician, but as I understand it, inclusion of variables in best-supported models is not in itself evidence that the variables are significant. Also, the world is shifting from a "best model" approach to a model-averaged approached (see Arnold, 2010; Doherty et al., 2012). It would not take much to calculate model-averaged parameter estimates and the results would be more robust for it. Finally, the authors could report cumulative AIC weights and/or new β values for each variable. Β values with CIs would of course inform readers about the statistical significance of season and migratory form as predictors.

Note that we are not claiming significance based on AIC – we are extremely careful not to use the word “significance” in the MS. While we appreciate the comment in general, we think that model averaging is not a universal remedy: there are many (including us) who suggest the approach is to explain what the common features of all well-supported models are, and to discuss how these models differ — the differences point to factors that may, or may not, be important, while the robust patterns hold well. Unfortunately, model averaging has problems of its own (Galipaud et al., 2014).

We have followed the advice of Burnham and Anderson, 2010 and also Arnold, 2010 (and informed by Galipaud et al.); averaging across model parameters is generally useful "if one has a large number of closely related models, such as in linear-regression based variable selection, designation of a single best model is unsatisfactory because that "best" model is often highly variable…" In our case model 1 has 60% of all the weight, while model 2 achieves 36%. The rest of the models carry very little weight. Both models agree on a number of key points, and only differ on whether sex-specificity also impacts the results. We therefore judged it to be robust to discuss the results on the model which is best supported, while reminding the reader of the properties of model 2, as well as some key recommendations of the AIC world:

“Our second model, which included sex and had modest support (model 2, δ AICc = 0.95, Table 1), predicted that migrants have higher winter survival probability than residents, which was also predicted by model 1. Sex differences were not substantial (during summer: male residents Φ = 0.90; 95% CI = 0.83 – 0.95; female residents Φ = 0.89; 95% CI = 0.89 – 0.94; male migrants Φ = 0.95; 95% CI = 0.89 – 0.98, female migrants Φ = 0.94; 95% CI = 0.87- 0.97; during winter: male residents Φ = 0.58; 95% CI = 0.51 – 0.65; female residents Φ = 0.53; 95% CI = 0.44 – 0.62; male migrants Φ = 0.75; 95% CI = 0.63 – 0.84, and female migrants Φ = 0.71; 95% CI = 0.60 – 0.81; detection probability was lower for migrants (P = 0.19; 95% CI = 0.13 – 0.26) than for residents (P = 0.74; 95% CI = 0.69 – 0.78)). It is reassuring that both models 1 and 2 agree on the importance of residency vs. migration in winter, while we refrain from making strong statements regarding the effect of sex, given that Burnham and Anderson 2002 (Burnham and Anderson, 2002) advise against considering inferior models competitive in cases like our model 2 (δ AIC within about 0-2 units of the best model, the difference being caused by 1 parameter added to the best model and the log-likelihood essentially unchanged).” Results and Discussion section.

We furthermore included AIC weights in the Table 1.

The subject of the paper is a bird, and as written the paper is rather bird-centric (or at least vertebrate oriented) in its approach, with the reference list dominated by vertebrate studies. To be relevant to the entire field of animal migration research, the authors could make some efforts to widen the scope. For example, –in the Introduction section, the authors state that migration can only evolve if migrants have a net survival benefit compared to residents, but it seems to me that this is a vertebrate phenomenon. What about multi-generational migrants (such as insects) where multiple migrations and breeding bouts occur throughout the year (including the "winter"). In a partially migrant insect system for example, migration could evolve if migrant lineages produce more progeny over the course of several generations of winter-breeding than resident lineages which survive winter by diapausing (and therefore not breeding). I would suggest broadening the introduction so that it covers all taxa will increase the impact of this paper.

As mentioned previously, we have broadened the Introduction section to include other taxa and an example of a noctuid moth which performs multigenerational migrations. We included the following sentence:

“Similarly, the noctuid moth (Autographa γ) performs a multi-generational migration which confer substantial reproductive benefits by allowing a lineage to spread to multiple sites (Chapman et al., 2012).”

Results and Discussion section: What is known about the reproductive performance of residents, is there any evidence that it is around 60% higher than migrants as predicted? Unless it is significantly higher, then migration should be favoured and expected to outcompete residency, but the number of migrants and residents in your study indicates that residency outnumbers migration by 2:1.

Unfortunately, we do not have data on breeding performance for either group. In cormorants (obviously a very different kind of bird), a new paper — published since we submitted the first version of our MS — shows that there are clear breeding success benefits of residency (Grist et al., 2017), though they do not reach anything as high as our 60% (the figures are 16-18% depending on sex). However, Grist et al. only compared birds that were breeding (thus ignoring any effect that a migrant might fail to occupy a breeding site), which means that the breeding benefits of residency may be underestimated, and cormorants only breed once per season; if we consider that blackbirds is a multi-brood species (up to 2-3 broods a year), then anything that extends potential breeding times to earlier could also allow, probabilistically, multiple brooding or replacement broods, which would further enhance success differences. We now write:

“We conclude from this calculation that the reproductive performance of residents would have to be 61.25% higher than in migrants to achieve equal fitness of the alternative strategies. Such benefits could come about from prior residency effects (either occupying a better territory or avoiding floater status), combined with a longer time spent in the breeding habitat which can make multiple nesting or re-nesting (in case of failure) more likely. Considering that blackbirds are a multi-brood species (2-3 broods a year), it could be possible that residents gain a 61.25% higher breeding success compared to migrants. Future studies need to confirm these calculations. If resident breeding success is higher than 61.25%, then the fitness of migrants will be lower than the fitness of residents and migration would be a conditional strategy operating under frequency-dependent selection. For conditional migration strategies, some intrinsic phenotypic characteristics (sex, age, dominance) result in a need to adopt a strategy that might yield overall lower fitness than what residents on average achieve, but it is the better choice to optimize individual fitness (Lundberg, 1987). To distinguish between these two alternatives, data of reproductive success for this species are needed (note that comparisons within existing studies, such as Grist et al., 2017 on cormorants, do not incorporate all the processes we have envisaged above).” Results and Discussion section.

References: The formatting is all over the place here.

Thank you. We carefully corrected the formatting of the References.

Figure 2: The legend indicates 22 birds were tracked, but plot A appears to have 28 points, while plot B appears to have data from 26 birds.

We carefully checked inconsistencies in Figure 2. The fact that we reported 22 birds but there are 29 points in Figure 2 is due to the fact that for 6 birds we had repeated measurements in different years. In the current version of the manuscript we represented this fact with different symbols in the figure. Regarding the inconsistency in Figure 2 in the previous version there are less observations because the x axis was truncated (due to a mistake in the code). In the current version we solved this issue.

Figure 3: The bars are yellow not orange.

Thank you very much for this comment. We replaced the word “orange for “yellow” in the Figure 3 legend.

Results and Discussion section: I'm not sure how these confidence intervals relate to the probability values, as the detection probability for migrants is given as 0.19 but has confidence intervals of 0.69 – 0.78.

Thank you very much for noticing our mistake. In the previous version, we reported the confidence intervals for the detection probability of migrants incorrectly. We replaced the sentence “taking into account the lower detection probability for migrants (P = 0.19; 95% CI = 0.69 – 0.78, Figure 3)” for “taking into account the lower detection probability for migrants (P = 0.19; 95% CI = 0.13 – 0.26, Figure 3)” Results and Discussion section.

Materials and methods section: Repeated text here.

We appreciate the reviewer caught this mistake. We deleted the repeated text “Juvenile birds were sexed using DNA-based sex identification”.

Introduction: "is necessary" I find this an over-generalisation of points that might apply to the system the authors have on mind. Many alternatives are possible, for example that later-returning migrants arrive in better state, with high reward, as proposed for example by Drent et al., 2003).

We fully agree with the reviewer comment and we changed the sentence “Superior winter survival, in models of partial migration (6), is necessary to compensate for the potential cost experienced by migrants of foregoing priority access to breeding resources.”

to read “Based on theoretical models of partial migration in birds (Kokko, 2011), which assume that residency offers reproductive benefits (access to better breeding territories) and that migration should confer survival benefits for at least some individuals if the winter conditions at the breeding ground are harsh, we predicted that migrants should have higher survival probabilities during the winter period, whereas summer survival might not differ between migrants and residents. Introduction section.

Furthermore, we added in the Discussion section the alternative explanation proposed by Drent et al., 2003:

“Furthermore, if residents that do survive harsh winters begin the breeding season in poor condition, then physically dominant migrants could successfully take-over prime territories from residents. Under this scenario, the prior residency effect would not be acting at full strength (Drent et al., 2003; Jahn et al., 2010; Kokko, 2011) and migrants would gain breeding benefits.”